# Co-Pyrolysis of Beet Pulp and Defecation Lime in TG-MS System

**Radosław Slezak [1],\*** , **Liliana Krzystek [1]** , **Piotr Dziugan [2]** and **Stanisław Ledakowicz [1]**

[1] Department of Bioprocess Engineering, Faculty of Process and Environmental Engineering, Lodz University of Technology, Wolczanska 213, 90-924 Lodz, Poland; liliana.krzystek@p.lodz.pl (L.K.); stanleda@p.lodz.pl (S.L.)

[2] Institute of Fermentation Technology and Microbiology, Faculty of Biotechnology and Food Sciences, Lodz University of Technology, Wolczanska 171-173, 90-924 Lodz, Poland; piotr.dziugan@p.lodz.pl

\* Correspondence: radoslaw.slezak@p.lodz.pl

**Abstract:** The process of pyrolysis of beet pulp, a by-product after the extraction of raw sugar from sugar beet, with the addition of defecation lime was studied in a thermobalance coupled with a mass spectrometer. The beet pulp pyrolysis process took place completely at 600 °C, and the resulting char, tar and gas were characterized by higher heating values of 23.9, 21.6 and 7.77 MJ/kg, respectively. The addition of the defecation lime to beet pulp caused both an increase in the char production yield and a decrease in the tar production yield. At the same time, the higher heating value of char and tar decreased along with the increase of defecation lime added to the sample. The deconvolution of derivative thermogravimetric (DTG) curves allowed us to identify the basic components of beet pulp, for which the activation energy by isoconversion method was calculated. The 20 wt.% addition of defecation lime caused an increase of the activation energy by about 18%. Further increase in the defecation lime content resulted in a reduction of activation energy. At the temperature above 600 °C, calcination of calcium carbonate contained in defecation lime occurred. The $CO_2$ produced during calcination process did not cause auto-gasification of char.

**Keywords:** beet pulp; defecation lime; pyrolysis; kinetics; char; tar; flammable gasses

## 1. Introduction

In production processes, there is continuing discussion about "circular economy", which involves creating a closed-loop system. Waste in such a process is treated as a substrate for the subsequent processes. The production of fuels, chemicals and energy from waste reduces dependence on fossil fuels and facilitates sustainable waste management [1]. The process of transforming waste into valuable products can be carried out using biochemical and thermochemical processes. Thermochemical processes are shorter and almost completely utilize biomass components. They are less sensible for changeable biomass compositions and do not require careful control of microorganism growth conditions in the bioreactor. Biochemical processes need more space to perform the pretreatments of biomass [2]. Among all thermochemical processes, only the pyrolysis process allows one to obtain products (char and tar) that can be stored for the longest time and easily transported. The pyrolysis process enables one to utilize the biomass properly and to divert the end-products' yield according to the desired outcome [3]. Considering capital and operating costs, greenhouse gas emissions, consumption of process chemicals and carbon efficiency, thermochemical processes with an indication of the pyrolysis process are more effective [4,5]. Carrying out the slow pyrolysis process is advantageous due to the small amount of formed tar [6,7], which is highly oxygenated, and complexed oil showing physico-chemical instability [8]. Char produced from lignocellulosic biomass and agro-industrial

waste is suitable for solid fuel production and particularly poultry litter char is suitable for soil amendment [3,9].

The EU is the world's leading producer of sugar beet, with approximately 50% of the global production; almost 21 million tons of beet sugar in 2017/2018 and 2.3 million tons in Poland. The sugar beet industry plays a critical part in the European agricultural economy. During beet sugar production, three types of wastes can be identified, namely the beet pulp, molasses and defecation lime. From 1 tonne of harvested beet, about 150 kg of sugar, 500 kg of beet pulp (with a dry matter content of 10 wt.% (weight percent)), 50 kg of defecation lime and 35 kg of molasses are produced [8]. Defecation lime is used for growing plants due to the high content of lime ($CaCO_3$) and the presence of $P_2O_5$, $K_2O$, $MgO$ and $Na_2O$. Molasses contains a lot of sugars and minerals that allow its use mainly in the fermentation industry (production of yeasts, alcohol, organic acids), as well as in the fodder and pharmaceutical industries [8].

The beet pulp is used as animal feed or as a raw material for methane production. The cosmetic sector uses the beet pulp as surfactant, and in the paper industry, compacted or dried beet pulp is used as a partial substituent of wood fibers [8]. Due to the decreasing interest of agriculture and the low price of electricity produced from biogas, new methods of managing this biomass are being sought in Poland. In addition, a new market situation related to sugar quotas abolition appeared in Poland in 2017. The sugar industry will have to meet market expectations and find new solutions to improve financial performance by using waste products as a substrate for the production of biofuels, through thermochemical processing for example.

Recently, an increasing number of publications on thermochemical valorization of the sugar industry waste can be observed in the topical literature, which have been summarized in the review article by Nicodème et al. [8]. Most of the research on sugar industry waste concerned the sugarcane bagasse [10–14] because the sugarcane bagasse has twice the cellulose content and more than 10 times higher lignin content than beet pulp [8]. To date, the pyrolysis process with sugar beet pulp has been studied in the thermobalance [15,16], in the fixed bed reactor [17,18] and in the fluidized bed reactor [19].

The second waste product, which, besides the beet pulp, was decided to be used in the co-pyrolysis process was the defecation lime. To date, lignite was the additive to the pulp used in the pyrolysis process. This process was investigated in the thermobalance [20] and in the fixed bed reactor [21]. The effect of the $CaCO_3$ addition on the co-pyrolysis process was studied with the sewage sludge [22,23] and biomass pellets [24]. Calcium carbonate only slightly influenced the decomposition rate of biomass pellets at temperatures below 400 °C [24]. At temperatures above 700 °C, calcium carbonate calcination takes place, resulting in $CaO$ and $CO_2$ production. Released $CO_2$ during calcining may cause auto-oxidation of the char [25]. The resulting $CaO$ can be reused as a basic ingredient in the production of limewash. This allows the $CaO/CaCO_3$ cycle to close, and the energy needed for $CaCO_3$ regeneration can be obtained from the products of beet pulp pyrolysis.

A description of the kinetics of the beet pulp pyrolysis process has been presented so far by Devrim [20] exploiting the Arrhenius model-fitting method. The model-fitting methods are often used, but the fact that they are only suitable for describing the one-step reaction mechanism is the significant limitation [26]. For better understanding of the kinetics of beet pulp pyrolysis, the isoconversional model, which gives better results than the model-fitting methods [27], was used in this work. The literature on the subject lacks the kinetics of co-pyrolysis for beet pulp with defecation lime. So far, the kinetics study on the co-pyrolysis process has been carried out for the beet pulp only with lignite [20]. There is also no information on the strict composition of the gas formed during pyrolysis of the beet pulp. Aho et al. [19] in their work presented only the changes in CO and $CO_2$ concentration during the produced gas in the process of beet pulp pyrolysis.

The low interest of farmers, as well as the seasonality of sugar production, is increasingly forcing sugar factories to dry beet pulp. Due to the high moisture content and the presence of organic matter, raw beet pulp cannot be stored for a long time because of biodegradation. Drying of sugar

beet pulp with initial solid percentage of around 10 wt.% is a highly energy-consuming process. However, the drying process of sugar beet pulp is well described in topical literature supported by industrial practice [28,29]. This process uses waste heat and thus offers a significant reduction in energy consumption. Moreover, the pyrolysis gas could be utilized for the drying process. Therefore, this paper focuses only on the pyrolysis process of dried sugar beet pulp.

The aim of the study was to determine the impact of defecation lime addition on char, tar and gas production yield in the pyrolysis of beet pulp from the sugar industry. The calorific value of the resulting products and kinetic constants of the pyrolysis process were calculated. A deconvolution of DTG (derivative thermogravimetric) curves was carried out to determine the effect of defecation lime on the auto-gasification process.

## 2. Materials and Methods

### 2.1. Substrate

In the conducted research, the beet pulp and defecation lime from sugar company Dobrzelin in Poland were used as substrates. The raw materials were first dried at 105 °C over 24 h and then ground in the Pulverisette 15 mill Fritzsch with 0.25 mm sieve. To investigate the effect of an inorganic compound on beet pulp pyrolysis, defecation lime was blended with beet pulp at mass fractions ($w$) equal to 0.2, 0.4, 0.6 and 0.8. The given $w$ refers to the content of defecation lime in the whole sample. For the sample containing only beet pulp, the $w$ parameter was equal to 0, whereas for defecation lime it was equal to 1. Proximate and elemental analyses of the beet pulp and defecation lime are presented in Table 1.

**Table 1.** Proximate and elemental analyses of dried beet pulp and defecation lime (average value ± standard deviation of three samples).

| Fraction | Beet Pulp ($w$ = 0 Mass Fraction) | Defecation Lime ($w$ = 1 Mass Fraction) |
|---|---|---|
| Proximate analysis | | |
| Moisture (wt.%) | 3.64 ± 0.08 | 0.52 ± 0.02 |
| Volatiles (wt.%) | 70.89 ± 0.91 | 46.94 ± 0.67 |
| Fixed carbon (wt.%) | 18.94 ± 0.29 | 0.13 ± 0.01 |
| Ash (wt.%) | 6.53 ± 0.12 | 52.41 ± 0.73 |
| Ultimate analysis | | |
| C (wt.%) | 40.47 ± 0.83 | 15.06 ± 0.22 |
| H (wt.%) | 5.84 ± 0.12 | 0.83 ± 0.04 |
| N (wt.%) | 0.77 ± 0.05 | 0.02 ± 0.01 |

### 2.2. Experimental

Experiments were performed in the thermobalance (TGA/SDTA 851e LF, Mettler-Toledo, Zürich, Switzerland) coupled with the mass spectrometer (MS) (QMS 200 Balzers Thermostar, Asslar, Germany). The connection between this thermobalance and MS was described in our earlier work [30]. In MS, the non-condensable gases produced in the highest quantities in the pyrolysis process ($H_2$, $CH_4$, CO, $CO_2$) and the water vapor were measured [31]. The experiments were carried out with dried sample of 20.0 ± 0.1 mg, which was placed into an aluminum crucible (150 μL) without the lid. A small mass of the sample in a crucible allowed us to reduce the possible occurrence of secondary reactions and thus not to alter the processes of mass and heat transfer [32]. The furnace was heated from 30 to 900 °C at rates equal to 5, 10 and 20 °C/min. The thermocouple (Pt–Rh10/Pt) to measure the sample temperature was placed directly under the sample holder. The argon (purity 99.999%) flowed through the furnace with the rate equal to 100 mL min$^{-1}$ to arrange for inert atmosphere in it. Each experiment was performed in triplicate, and the arithmetic mean was taken for data interpretation. The experimental

error of mass loss, elemental composition and temperature measurement was below 1.5%. Gas composition was measured with an error not exceeding 5.0%. The char samples for ultimate analysis were taken when the furnace reached the set temperature and then cooled to room temperature.

*2.3. Analytical Methods*

The moisture, volatiles, fixed carbon and ash content were determined using thermobalance TGA/SDTA 851e (Mettler-Toledo, Zürich, Switzerland) as the error of thermogravimetric method is lower than 6% [33]. The content of C, H and N in the sample was determined using elemental analyzer NA 2500 (CE Instruments, Hindley Green, UK). All analytical procedures were carried out according to standard methods [34]. Calibration of MS was performed in accordance with the method proposed by Nowicki and Ledakowicz [30]. The deconvolution of the DTG curve was performed using the Peak Sample Separation software from NETZSCH (Selb, Germany). The calculation of kinetic parameters was performed in the Kinetics program of the Netzsch Company (Selb, Germany). The higher heating value (HHV) of the substrate and char was calculated according to Dulong formula (Equation (1)) [35].

$$HHV = 0.3383C + 1.422\left(H - \left(\frac{O}{8}\right)\right) \tag{1}$$

where C, H and O are respectively the mass fraction of carbon, hydrogen and oxygen in the sample. Oxygen content in the samples was determined using the method described by Monlau et al. [36].

## 3. Results and Discussion

*3.1. Kinetic Analysis*

The beet pulp is mainly composed of carbohydrates such as pectin, hemicellulose, cellulose and lignin, whereas defecation lime consists mostly of calcium bicarbonate. In Figure 1a showing the distribution of beet pulp and defecation lime in the pyrolysis process (DTG curves), three phases can be distinguished. The weight loss in the first phase (30–180 °C) was primarily associated with dehydration of sample, which was confirmed by gas analysis in MS (Section 3.2). In the temperature range 180–600 °C the pyrolysis process took place (phase II) during which char, tar and gas were formed. Above 600 °C (phase III), decomposition of $CaCO_3$ with $CO_2$ evolution took place, which could cause auto-gasification of the char formed in phase II.

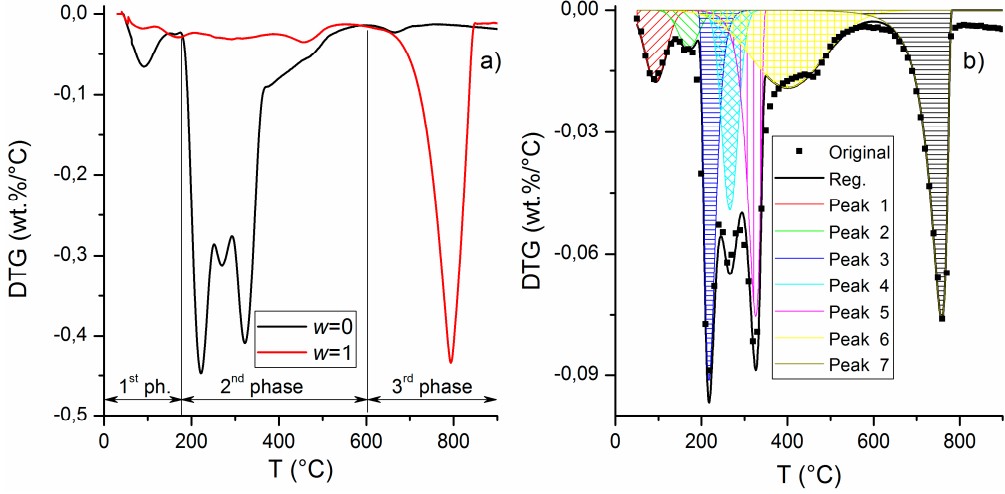

**Figure 1.** Derivative thermogravimetric (DTG) curves for pyrolysis of samples with mass fraction equal to 0 and 1 (**a**) and deconvolution of DTG curve for mass fraction of defecation lime in sample equal to 0.4 (**b**) at heating rate of 10 °C/min.

Figure 1b shows the deconvolution of the DTG curve for the sample with a mass fraction of defecation lime equal to 0.4 for the furnace heating rate of 10 °C/min. The deconvolution of DTG curves is often used to represent the distribution of individual compounds in a sample [37]. The experimental data were fitted as an additive superposition of peaks, where each single peak was represented by the Fraser–Suzuki profile. Each peak was characterized by the parameter presented in Table 2. The best fit to the DTG curve using the least number of profiles was achieved for a system consisting of seven peaks (Figure 1b).

**Table 2.** Deconvolution parameters of the DTG curve for mass fraction of defecation lime in the sample equal to 0.4.

| Phase | | I | | II | | | | III |
|---|---|---|---|---|---|---|---|---|
| Peak No. | | 1 | 2 | 3 | 4 | 5 | 6 | 7 |
| **Position** (°C) | | 93.2 | 170.0 | 217.3 | 266.0 | 326.0 | 400.0 | 758.2 |
| **Amplitude** (wt.%/°C) | | −0.0176 | −0.00922 | −0.0911 | −0.0493 | −0.0754 | $-0.0191^2$ | −0.00760 |
| **Halfwidth** (°C) | | 56.8 | 55.4 | 29.5 | 44.4 | 33.4 | 199.7 | 42.5 |
| **Asymmetry** - | | 0 | 0 | −0.40 | 0 | 0.60 | 0 | 0.84 |

In phase I, two peaks (No. 1 and 2) were observed, which were associated with bulk and bound water, respectively. In phase II, in which the pyrolysis process took place, four peaks were identified (No. 3–6). The maximum decomposition rate for the peak No. 3 occurred at 217 °C, and it was associated with pectin degradation. The next peak (No. 4) described the decomposition of hemicellulose, with the maximum decomposition rate at 266 °C. At 326 °C, the maximum decomposition rate was related to cellulose decomposition (peak No. 5), whereas peak No. 6 was responsible for the decomposition of lignin. The extracted peaks coincide with the peaks for individual biomass components (hemicellulose, cellulose and lignin) obtained by Yang et al. [31] in the pyrolysis process. The DTG curves during pyrolysis of beet pulp in the thermobalance in our studies are similar to the results obtained by Sidi-Yacoub et al. [15], Simkovic et al. [16] and Yilgin et al. [21].

At temperatures above 600 °C (phase III), there was a decomposition of $CaCO_3$ contained in defecation lime and probably char decomposition as a result of reverse Boudouard reaction [8]. The reactions taking place above the temperature of 600 °C are presented by the peak No. 7. In order to check whether the gasification process of the char took place in phase III, additional calculations were carried out. Based on the surface area of peak No. 7 for the sample containing defecation lime, the amount of $CaCO_3$ decomposed was calculated. Then, for samples with mass fractions of defecation lime from 0.2 to 0.8, the theoretical area of peak No. 7 connected only with decomposition of $CaCO_3$ was calculated. Comparing the theoretical and actual areas of peak No. 7, it was observed that the difference between the values was below 5%. The obtained results from comparison to the area of peak No. 7 show that at a temperature above 600 °C, there was no significant auto-gasification of the resulting char. In our experiments, the auto-gasification process did not take place because it was carried out in a thermobalance, in which the $CO_2$ concentration was very low and the residence time of the gas was very short. But in the pyrolysis reactors, the $CO_2$ concentration is significantly higher and the gas residence time is much longer. Such conditions favor the auto-gasification process, although the rate of gasification is significantly lower than the one of the pyrolysis process.

The Friedman's isoconversion method was used to calculate the activation energy (Equation (2)).

$$\ln\left(\beta\frac{d\alpha}{dT}\right) = \ln(k_0)\frac{E}{R \cdot T} + \ln[f(\alpha)] \tag{2}$$

$$\alpha = \frac{m_0 - m_\tau}{m_0 - m_f} \tag{3}$$

where $\beta$ is the furnace heating rate (K/min), $\alpha$ is the degree of conversion described by Equation (3) (-), $T$ is the furnace temperature (K), $k_0$ stands for the pre-exponential factor (1/s), $E$ quantifies the energy of activation (kJ/mol), R is the universal gas constant 8.314 J/(mol·K), $f(\alpha)$ is the differential form of the

reaction model, $m_0$ is the initial mass of sample (g), $m_\tau$ is the mass at time $\tau$ (g) and $m_f$ is the final mass at the end of pyrolysis (g).

Only peaks No. 3, 4 and 5 were selected for further analysis due to a 70% share in mass loss during beet pulp pyrolysis. The degree of conversion for peaks No. 3, 4 and 5 and the mass fraction of defecation lime in the range of 0 to 0.8 are shown in Figure 2a. No data are presented for the test with $w = 1$ (defecation lime) because peaks No. 3, 4 and 5 were not isolated during the deconvolution of the DTG curve. The lack of peaks was likely caused by the very low content of organic matter in defecation lime. Increasing the amount of added defecation lime resulted in a decrease in the conversion degree of peaks 3, 4 and 5 due to the increase in the area of peak No. 7 being responsible for the decomposition of defecation lime. Changes in the activation energy determined by the Friedman method for peaks No. 3, 4 and 5 are shown in Figure 2b. The activation energy corresponds to the minimum energy required for breaking the chemical bonds between atoms and beginning the reaction. The higher value of the activation energy means that the reaction is slower and more difficult to initiate [38]. The activation energy for a given peak was read for a given conversion degree of the sample which corresponded to the maximum height of a given peak (Figure 2a).

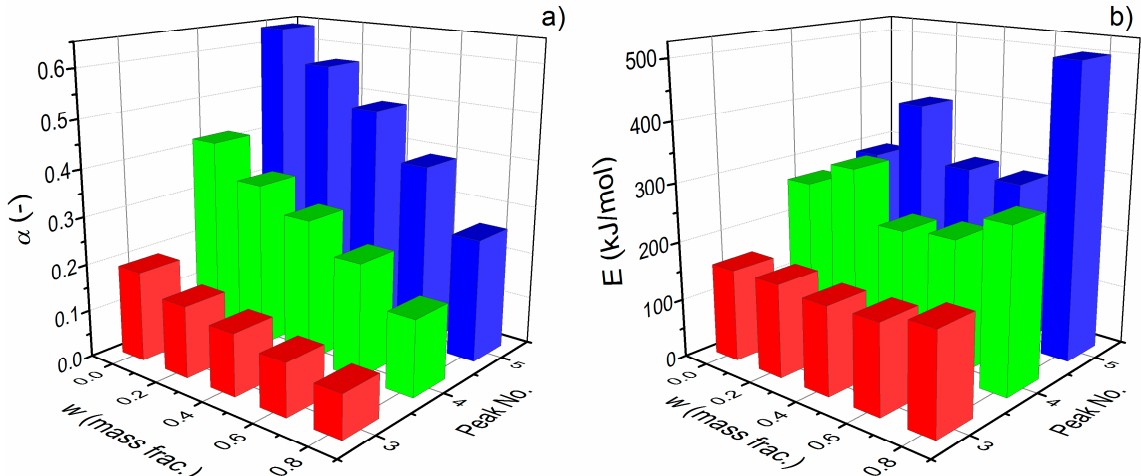

**Figure 2.** Changes of conversion degree (**a**) and energy activation (**b**) for peak No. 3, 4 and 5 at different ratios of mass fraction of beet pulp for an oven heating rate of 10 °C/min.

The activation energy values for pectin, hemicellulose and cellulose for beet pulp ($w = 0$) were 138, 267 and 287 kJ/mol, respectively. At the beginning of the pyrolysis process, compounds that are weakly linked to the linear polymeric chain break down, and therefore the decomposition of pectin and hemicellulose requires low energy. Then, random scission of linear chains occurs, which leads to an increase in activation energy of cellulose [39]. In the subject literature, the activation energy for pectin, hemicellulose and cellulose was respectively in the range 100–110 kJ/mol [40], 90–225 kJ/mol and 145–285 kJ/mol [41]. The addition of 0.2 mass fraction of defecation lime to beet pulp increased the activation energy by about 18%, which indicates the negative impact of defecation lime on the pyrolysis process of the beet pulp. Further increasing the mass fraction of defecation lime in the sample to a value of 0.4 resulted in a reduction of activation energy by about 6% relative to the activation energy value for the beet pulp. Devrim [20] obtained a similar relation during co-pyrolysis of sugar beet pulp with lignite. An increase in lignite content in the sample from 0 to 0.4 mass fraction caused a reduction of activation energy by 31%. The differences between our results and those of other research results from different measurement methods and kinetic models that were used. In addition, the preparation of the sample for analysis and its origin could also have an impact.

### 3.2. Production Yield of Char, Tar and Gas

As a result of the pyrolysis process, three products are formed in different physical states: solid (char), liquid (tar with water) and non-condensable gases. An exemplary profile of formed gases is presented for pyrolysis of beet pulp (Figure 3). By analyzing the gas profile, it can be observed that mainly $CO_2$ and CO are formed in the second phase. The production of these gases is largely related to the decomposition of hemicellulose and cellulose. The increase in methane production at 500 °C was mainly associated with lignin degradation [42]. Above 600 °C (phase III) $H_2$, CO and $CO_2$ were produced, which was associated with partial gasification of the char. During the gas analysis on MS, other gases were observed, such as $C_2H_6$, $C_3H_8$, $C_4H_{10}$, $CH_3OH$, $C_2H_5OH$, $C_6H_6$, $SO_2$ and $H_2S$; however, their quantification is very difficult. The gas components measured in this work ($H_2$, $CH_4$, CO, $CO_2$) constitute over 92% of all non-condensing gases formed in the pyrolysis process [42,43]. Mineral compounds added to biomass have an impact on the yield of pyrolysis products. The inorganic constituents promote secondary reactions which cause the breakdown of higher molecular compounds to smaller ones [44]. Due to the lack of significant auto-gasification of the char, the yields of the pyrolysis process were determined at a temperature of 600 °C. The obtained yields of pyrolysis products were referred to dry basis (Figure 4). In the studies conducted by Sklyadnev et al. [18], the temperature of 510 °C was assumed to be the optimal temperature for beet pulp pyrolysis. This work presents the yields of the pyrolysis process only for samples with a mass fraction of defecation lime in the range from 0 to 0.4, due to low HHV of pyrolysis products with a higher mass fraction of defecation lime. A detailed description of HHV calculated for substrates and pyrolysis products is provided in Section 3.4.

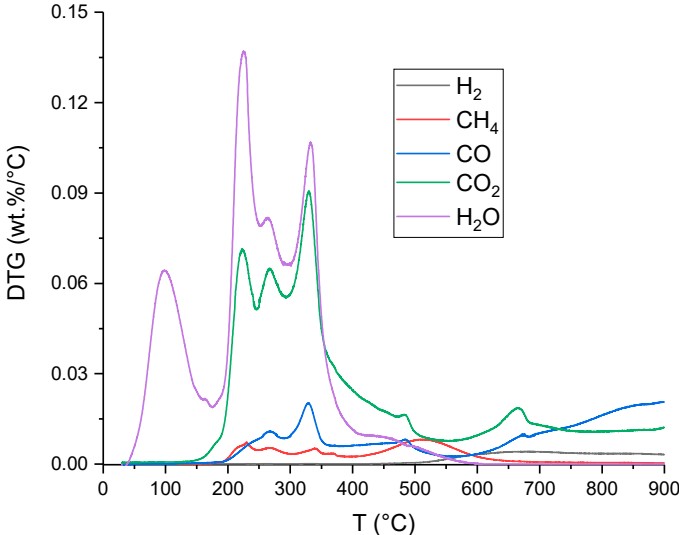

**Figure 3.** The profile of evolved gases during pyrolysis of beet pulp.

The increase in the mass fraction of defecation lime in the sample from 0 to 0.4 resulted in an increase in the yield of char production from 29.4 to 51.6 wt.%. This increase was mainly due to the higher ash content in the sample. However, increasing the mass fraction of defecation lime in the beet pulp from 0 to 0.4 resulted in a reduction of the tar and gas production yield by 49 and 32 wt.%, respectively. In the research of Ranzi et al. [45], during rapid pyrolysis of biomass, it has also been observed that a higher content of mineral compounds causes a decrease in the yield of the bio-oil production. The concentration of defecation lime in the sample had no significant effect on the water production yield, which was approximately 21 wt.%. The $H_2O$ is a common product from pyrolysis of hemicelluloses, cellulose and lignin [46]. Comparison of the production yield with those obtained by other authors is difficult due to the fact that the pyrolysis process is influenced by particle size, sample type, pyrolyzer type and temperature. So far, in the research conducted by DeGroot et al. [17], regarding the slow pyrolysis of sugar beet pulp, the production yield for char was 21.2 wt.%. In the

fast pyrolysis process of beet pulp carried out by Aho et al. [19] and Yilgin et al. [21], the yield of char production was in the range of 17–25 wt.%, oil 36–38 wt.% and gas 21–40 wt.%.

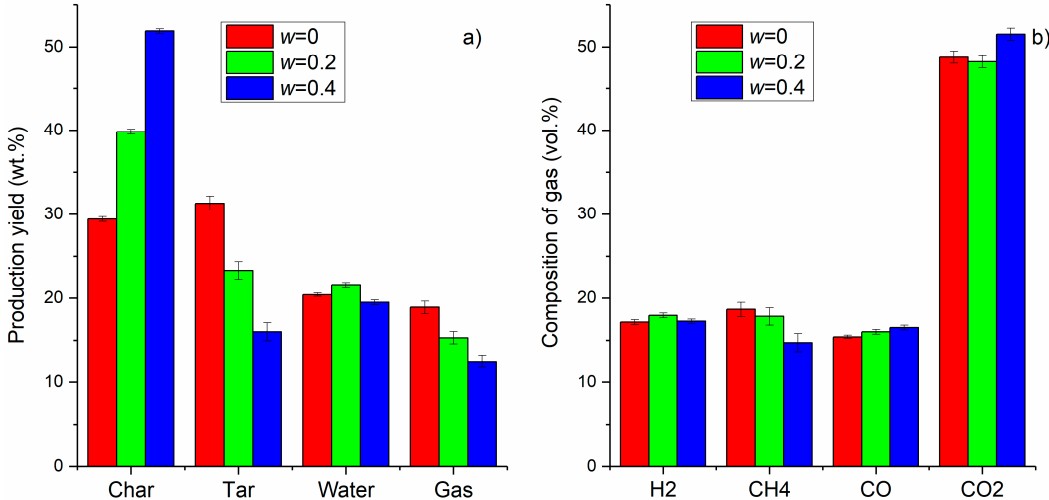

**Figure 4.** Production yields (**a**) and gas composition (**b**) during pyrolysis at 600 °C.

Defecation lime did not have a significant impact on the composition of gases ($H_2$, $CH_4$, CO and $CO_2$) produced during pyrolysis at 600 °C. The $CO_2$ content was twice as high as the content of other gases shown in Figure 4. Similarly, in studies by Aho et al. [19], during pyrolysis of beet pulp in the fluidized bed reactor, the $CO_2$ concentration was twice as high as CO. In these studies, the concentration of $H_2$ and $CH_4$ was approximately equal to 17 vol.%.

*3.3. Char Properties*

Content of H and O in char after pyrolysis process tends to be lower due to dehydration, demethanation, decarboxylation and decarbonylation of oligosaccharides. The higher the temperature of the pyrolysis process, the lower the H and O content in the char [47]. Removal of O caused an increase of char hydrophobicity. The reduction in O content is an effect of the removal of acidic functional groups, causing the char to become more alkaline [48]. Removing hydrogen from the sample results in unsaturated structures of char, which indicates the higher aromaticity of the char [49]. In contrast, the carbon content of the char after the pyrolysis process increases for most biomass [50]. However, for sewage sludge containing a large amount of ash, a reduction in carbon content has been observed [49]. Figure 5a presents changes in the content of elements C, H, N and O in the char after the beet pulp pyrolysis process in the temperature range of 30–900 °C. Analyzing the changes in the elemental composition of char from beet pulp for different pyrolysis temperatures, it was concluded that the largest changes in the composition occur in the temperature range of 200–600 °C. At 500 °C, the C content in the char was 64.7 wt.%. For comparison, in the work of Aho et al. [19], during pyrolysis of the beet pulp in a fluidized bed reactor at a temperature of 500 °C, the carbon content in the char was higher by 17.5% than it was in our research. However, in the studies of Sklyadnev et al. [18], during pyrolysis of sugar beet pulp in a fixed bed reactor at 510 °C, the carbon content in the char was equal to 85.6 wt.%.

The effect of the addition of defecation lime to the sample on the H/C and O/C ratio before and after the pyrolysis process is depicted in the Van Krevelen diagram (Figure 5b). Figure 5b shows the change of char composition before and after pyrolysis at 600 °C. Increasing mass fraction of defecation lime in the sample before the beginning of the process caused an increase in the O/C ratio and a decrease in the H/C ratio, which was due to the fact that defecation lime consists mainly of calcium carbonate. In the char after pyrolysis at 600 °C, it can be seen that the increase in the defecation lime content caused a significant increase in the O/C. The low value of the H/C and O/C ratio affects the HHV char value (Figure 6a–c).

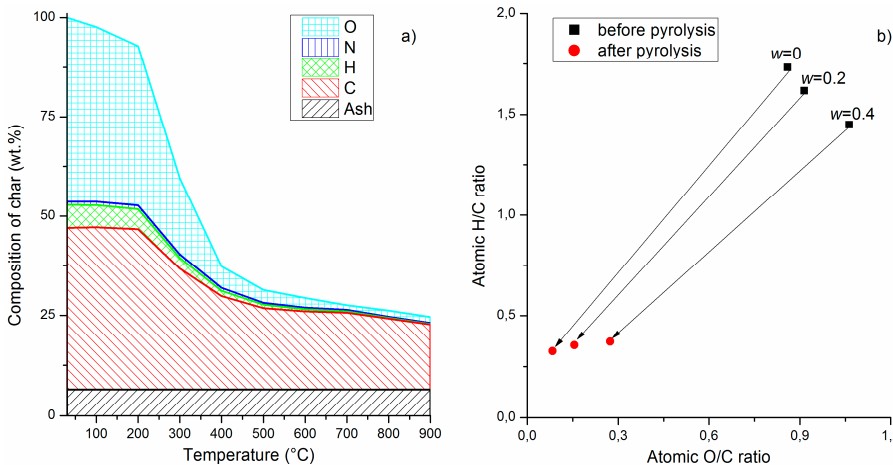

**Figure 5.** Composition of char after pyrolysis of beet pulp at different temperatures referred to initial weight of the sample (**a**) and changes of atomic H/C and O/C ratio before and after pyrolysis at 600 °C for mass fraction of defecation lime in sample from 0 to 0.4 (**b**).

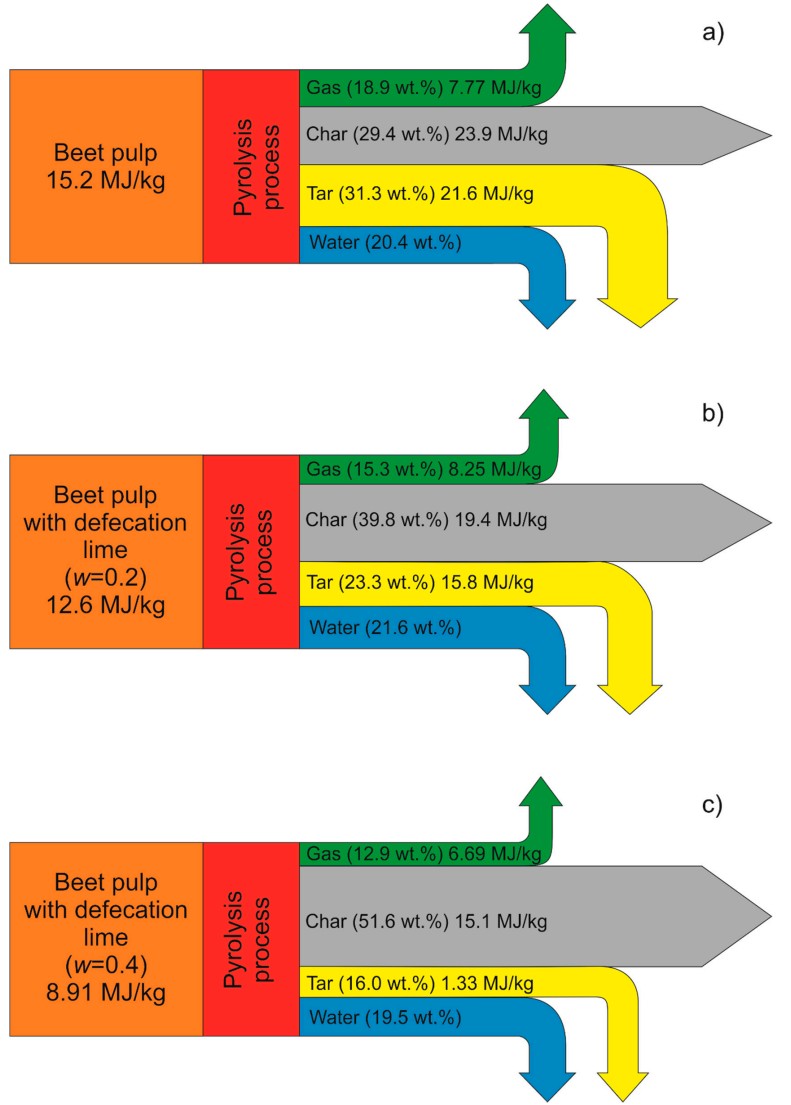

**Figure 6.** Sankey diagrams for pyrolysis process at temperature 600 °C of samples with mass fraction of defecation lime equal to 0 (**a**), 0.2 (**b**) and 0.4 (**c**).

*3.4. Energy Balance of Pyrolysis Process*

The low mass of the sample subjected to the process of pyrolysis in a thermobalance prevented the separation of the resulting tar and determination of its HHV. This value was calculated based on the energy balance, which assumed that the amount of energy contained in the substrate is equal to the energy in products after the pyrolysis process (Equation (4)).

$$HHV_{substarte} = HHV_{char} \cdot Y_{char} + HHV_{tar} \cdot Y_{tar} + HHV_{gas} \cdot Y_{gas} \qquad (4)$$

where HHV is the higher heating value (MJ/kg), and Y stands for the production yield (wt./wt.). The HHV of beet pulp was about 15 MJ/kg (Figure 6a).

Biomass with low lignin content such as food waste has HHV in the range from 17 to 19.5 MJ/kg [51,52]. For comparison, HHV of wood residue from a mixture of pine, oak and birch is about 18 MJ/kg [53]. The pyrolysis process caused an increase in HHV of the solid pyrolysis residue. The highest HHV value of char (23.9 MJ/kg) was observed in the beet pulp sample without the addition of defecation lime. From literature review, the HHV of char from food waste increases from 4 to 9 MJ/kg [54]. Similarly, in the work of Wang et al. [55], during the study of six different biomasses (peanut shell, maize cob, wheat straw, rice lemma, pine sawdust and bamboo sawdust), the average HHV value of char was $24.3 \pm 5.2$ MJ/kg. In this work, increasing the mass fraction of defecation lime in the sample resulted in a reduction in HHV value of both substrate and char. The addition of defecation lime to the beet pulp did not have a notable effect on the HHV of the resulting gas. Similarly, Yuan et al. [24] observed that the addition of $CaCO_3$ has no considerable effect on the gas composition during pyrolysis carried out below 600 °C. The obtained value of HHV gas (7.7 MJ/kg) produced in the pyrolysis process was similar to the values obtained by other researchers [56]. The HHV of tar for beet pulp calculated from the energy balance was equal to 21.6 MJ/kg (Figure 6a). For comparison, during pyrolysis of energy crops [57] and wood waste [58], tar was characterized by HHV in the range from 23 to 27 MJ/kg. In our research, the addition of defecation lime has reduced the HHV of tar. In the research of Kwon et al. [22], it has been observed that increasing the addition of $CaCO_3$ to the sludge in the pyrolysis process caused a reduction in the content of polycyclic aromatic hydrocarbons (PAHs), which are characterized by high HHV.

## 4. Conclusions

The beet pulp from the sugar industry is a good substrate for the pyrolysis process due to its chemical composition, which allows the process to be carried out below 600 °C. Received char and tar exhibited an increase of energy density. Low HHV of produced gases was caused by a 2-fold higher concentration of $CO_2$ than the one of $H_2$, $CH_4$ and CO. The addition of the defecation lime to beet pulp caused both an increase in activation energy of pyrolysis process and a decrease in the higher heating value of char and tar. Even though experiments, performed above 600 °C in thermobalance with the beet pulp doped with defecation lime, did not cause auto-gasification of char, this process can be expected in the pyrolysis reactor because of higher $CO_2$ concentration and longer residence time. This hypothesis needs further experimental verification.

**Author Contributions:** Conceptualization, R.S., P.D. and S.L.; methodology, R.S. and S.L.; validation, R.S. and S.L.; formal analysis, R.S. and S.L.; investigation, R.S.; writing, review and editing, R.S., L.K. and S.L.; supervision, P.D., L.K. and S.L. All authors have read and agreed to the published version of the manuscript.

**Funding:** This research received no external funding.

**Conflicts of Interest:** The authors declare no conflict of interest.

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
