# Peer review of "Co-Pyrolysis of Beet Pulp and Defecation Lime in TG-MS System"

_energies, doi:10.3390/en13092304_

Round 1

Reviewer 1 Report

The article reports a very interesting analysis of the potential of the copyrolysis of beet pulp and defecation lime. It is very well structured and all the "essential" aspects for the evaluation of the biomass quality at the pyrolysis process are sufficiently analyzed. If it were a chemical-oriented journal it would therefore be acceptable without major changes. Since it is proposed on Energies, it is necessary to take a step back. A biomass that initially has a solid percentage of around 10% cannot be used for a dry thermochemical process. It can at best be interesting for the production of Hydrochar and liquids from HTC. The drying process is in fact extremely energy-intensive. This is not to say that the article should not be published, but should convince more about why using such a wet biomass in a dry process. Maybe you can think of some integration in some production process. But it is a very circumstantial eventuality and should be very well evaluated. The energy balances at the end of the article are in fact very vague and not very sensible in this regard. The evaluation is therefore negative not on the article quality, but on the initial definition of the used biomass .

Reviewer 2 Report

The article is of scientific interest and should be suitable for publication; however this needs

minor revision, so that it can be accepted by the journal.

All my corrections are in the manuscript.

Please improve the table 1, headings, and the abbreviations in the text.

Reviewer 3 Report

Co-pyrolysis of beet pulp and defecation lime in TG3 MS system

The work deals about determine the impact of defecation lime addition on char, tar and 83 gas production yield in the pyrolysis of beet pulp from the sugar industry.

The topic ir not very novelty but it can be interesting, but.... I start by the end:

-Lines 306-309.

 “…the addition of defecation lime was not  justified due to the low calorific value of the resulting char and tar. Carrying the pyrolysis process of beet pulp with the addition of defecation lime above 600°C is not justified due to the lack of auto gasification of char by CO2 formed in the calcination process”.

I'm according with authors. The work has not an important conclussions. The addition of defecation lime was not justified. Then, is it a failed work?.

Could authors write a more atractive and interesting conclusions?.

Also, the most commun use for sugar beet molasses is fermentation for biogas production. Authors could explain the advantages of pyrolysis process vs fermentation in introduction section.

On the other hand, the article is rigorous, the methodology is appropriate and discussion is very well compared with other authors. The results are not very interesting but, in my opinion, the paper is good and summarizes various aspects about pyrolysis similar to a review article.

Some little questions or mistakes:

-Line 4: LEDAKOWICZ and???????

-Line 31: “bio- and….”

-Lines 48-19: “….in paper industry compressed or dried beet pulp is used 49 as an alternative component”. Is it complete the sentence?.

-Line 107. “…at the rate from 5 to 20ºC/min…”. Specific rates must be specified.

-Lines 116/17. “The moisture, volatiles, fixed carbon and ash content was determined using thermobalance 117 TGA/SDTA 851e (Mettler-Toledo, Zürich, Switzerland)”. Is it correct?.

-Line 123. Hhv was not measured in a calorimetric pump, I think the expression used from Dulong must be included in the text.

-Lines 302-306. These are results, not conclusions

.-Line 372. “2005” must be wrote in bold.

Then, article could be suitable for publication in “Energies” but authors must attend to comments about conclusions and advantages of pyrolysis.

Reviewer 4 Report

In general, the discussion of the results should be improved They should give a profile of the gases generated

Have the authors identified gases other than H2, CH4, CO, CO2?

They could add references in section 3.4. of raw materials similar in composition to beet pulp since the comparisons are made with materials that have higher content in lignin or with sewage sludge.

Could improve the quality of Figure 3

Reviewer 5 Report

The research is well defined and it is a topic very interesting for researchers in the field of biomass and from an energetic point of view. I recommend the mauscript for publication.

Author Response

We would like to thank this Reviewer for such opinion.

Round 2

Reviewer 1 Report

The paper can now be accepted in the present form. The explanations provided to all the reviewers requests, have surely improved the overall quality. 

If the authors consider them useful for the paper, i suggest them to read something of Fábio Codignole Luz regarding systems and modeling for the fast pyrolisys. 

Best regards

Author Response

We would like to thank the reviewer for valuable and constructive comments and suggestions.

Additional references have been added (page 2, line 42 and page 2, line 45)

  1. Codignole Luz, F.; Cordiner, S.; Manni, A.; Mulone, V.; Rocco, V. Biomass fast pyrolysis in screw reactors: Prediction of spent coffee grounds bio-oil production through a monodimensional model. Energy Convers. Manag. 2018, 168, 98–106.
  2. Codignole Luz, F.; Cordiner, S.; Manni, A.; Mulone, V.; Rocco, V. Biochar characteristics and early applications in anaerobic digestion-a review. J. Environ. Chem. Eng. 2018, 6, 2892–2909.

Reviewer 3 Report

The changes were attended. I think the article is suitable for publication in this revised form.

Author Response

We would like to thank the reviewer for valuable and constructive comments and suggestions.